# Safety and Non-Inferiority Evaluation of Two Immunization Schedules with an Inactivated SARS-CoV-2 Vaccine in Adults: A Randomized Clinical Trial

**DOI:** 10.3390/vaccines10071082

**Published:** 2022-07-06

**Authors:** Katia Abarca, Carolina Iturriaga, Marcela Urzúa, Nicole Le Corre, Augusto Pineda, Carolina Fernández, Angélica Domínguez, Pablo A. González, Susan M. Bueno, Paulina Donato, Pilar Espinoza, Daniela Fuentes, Marcela González, Paula Guzmán, Paula Muñoz-Venturelli, Carlos M. Pérez, Marcela Potin, Álvaro Rojas, José V. González-Aramundiz, Nicolás M. S. Gálvez, Francisca Aguirre-Boza, Sofía Aljaro, Luis Federico Bátiz, Yessica Campisto, Mariela Cepeda, Aarón Cortés, Sofía López, María Loreto Pérez, Andrea Schilling, Alexis M. Kalergis

**Affiliations:** 1Departamento de Enfermedades Infecciosas e Inmunología Pediátrica, División de Pediatría, School of Medicine, Pontificia Universidad Católica de Chile, Santiago 8330077, Chile; carolina.iturriaga@uc.cl (C.I.); maurzua@med.puc.cl (M.U.); n.lecorre.p@gmail.com (N.L.C.); anpineda@uc.cl (A.P.); fernandezcarola.2020@gmail.com (C.F.); mpotin@ucchristus.cl (M.P.); 2Millennium Institute on Immunology and Immunotherapy, Santiago 3871336, Chile; pagonzalez@bio.puc.cl (P.A.G.); sbueno@bio.puc.cl (S.M.B.); nrgalvez@uc.cl (N.M.S.G.); 3Departamento de Enfermedades Infecciosas del Adulto, División de Medicina Interna, School of Medicine, Pontificia Universidad Católica de Chile, Santiago 8330077, Chile; arojasgo@uc.cl; 4Medicina Física y Rehabilitación, Red de Salud UC Christus, Santiago 8320000, Chile; 5Departamento de Salud Pública, School of Medicine, Pontificia Universidad Católica Chile, Santiago 8330077, Chile; mndoming@uc.cl; 6Departamento de Genética Molecular y Microbiología, Facultad de Ciencias Biológicas, Pontificia Universidad Católica de Chile, Santiago 8330077, Chile; 7Complejo Asistencial Dr. Sótero del Río, Santiago 8150215, Chile; paudonatoi@gmail.com (P.D.); yycampis@uc.cl (Y.C.); 8Hospital Félix Bulnes, Facultad de Medicina y Ciencia y Facultad de Ciencias para el Cuidado de la Salud, Universidad San Sebastián, Santiago 8320000, Chile; pilar.espinoza@uss.cl (P.E.); carlos.perez@uss.cl (C.M.P.); perezlor@gmail.com (M.L.P.); 9Hospital Carlos Van Buren, Universidad de Valparaíso, Valparaíso 2340000, Chile; daniela.fuentes@uv.cl (D.F.); marielabeatrizcepeda@gmail.com (M.C.); 10Hospital Gustavo Fricke, Universidad de Valparaíso, Viña del Mar 2340000, Chile; marcela.gonzalez@uv.cl; 11Clínica Universidad de Los Andes, Servicio de Pediatría, Universidad de Los Andes, Santiago 8320000, Chile; pguzmanm@clinicauandes.cl; 12Facultad de Medicina Clínica Alemana, Universidad del Desarrollo, Santiago 8320000, Chile; paumunoz@udd.cl (P.M.-V.); dra.andrea.schilling@gmail.com (A.S.); 13Clínica San Carlos de Apoquindo, Red de Salud UC, Santiago 8320000, Chile; sofiaaljaro@gmail.com (S.A.); sulopez@miuandes.cl (S.L.); 14Departamento de Farmacia, Facultad de Química y de Farmacia, Pontificia Universidad Católica de Chile, Santiago 8330077, Chile; jvgonzal@uc.cl; 15Clínica Universidad de Los Andes, Unidad de Docencia, Investigación y Extensión Científica (DIDeP), Santiago 8320000, Chile; faguirre@clinicauandes.cl (F.A.-B.); acortes@clinicauandes.cl (A.C.); 16Escuela de Medicina, Facultad de Medicina, Universidad de los Andes, Santiago 8320000, Chile; lbatiz@uandes.cl; 17Centro de Investigación e Innovación Biomédica (CiiB), Universidad de los Andes, Santiago 8320000, Chile; 18Departamento de Endocrinología, Facultad de Medicina, Escuela de Medicina, Pontificia Universidad Católica de Chile, Santiago 8320000, Chile

**Keywords:** CoronaVac^®^, phase III clinical trial, SARS-CoV-2, COVID-19, vaccines, immunization schedules

## Abstract

Several vaccines have been developed to control the COVID-19 pandemic. CoronaVac^®^, an inactivated SARS-CoV-2 vaccine, has demonstrated safety and immunogenicity, preventing severe COVID-19 cases. We investigate the safety and non-inferiority of two immunization schedules of CoronaVac^®^ in a non-inferiority trial in healthy adults. A total of 2302 healthy adults were enrolled at 8 centers in Chile and randomly assigned to two vaccination schedules, receiving two doses with either 14 or 28 days between each. The primary safety and efficacy endpoints were solicited adverse events (AEs) within 7 days of each dose, and comparing the number of cases of SARS-CoV-2 infection 14 days after the second dose between the schedules, respectively. The most frequent local AE was pain at the injection site, which was less frequent in participants aged ≥60 years. Other local AEs were reported in less than 5% of participants. The most frequent systemic AEs were headache, fatigue, and myalgia. Most AEs were mild and transient. There were no significant differences for local and systemic AEs between schedules. A total of 58 COVID-19 cases were confirmed, and all but 2 of them were mild. No differences were observed in the proportion of COVID-19 cases between schedules. CoronaVac^®^ is safe, especially in ≥60-year-old participants. Both schedules protected against COVID-19 hospitalization.

## 1. Introduction

In March 2020, the COVID-19 pandemic, a disease caused by severe acute respiratory syndrome coronavirus 2 (SARS-CoV-2), was declared [1]. Two years into this pandemic, more than 250 million cases have been diagnosed worldwide, and more than 5 million deaths have been related to COVID-19 [2]. In Chile, since March 2020, 1.7 million laboratory-confirmed cases have been reported and more than 38,000 deaths have been related to COVID-19 as of December 2021 [3].

Initial COVID-19 outbreaks exhibited high morbidity and mortality in individuals over 60 years of age or with comorbidities, such as obesity, chronic pulmonary disease, cardiac disease, and the immunosuppressed population [4,5]. Antiviral drugs and immunomodulators have not been successful forms of treatment [6]. Prophylactic strategies with drugs such as hydroxychloroquine or ivermectin did not show any significant reduction in the risk of SARS-CoV-2 infection [7]. Other treatments, such as post-exposure type I interferon prophylaxis, are still being evaluated [8].

Vaccination is an essential prophylactic strategy to prevent pathogen spreading and disease caused by viral infection [9]. Early in the pandemic, the development of vaccines against SARS-CoV-2 was vigorously pursued. Different vaccine platforms were generated to prevent COVID-19, such as mRNA vaccines and viral vector-based vaccines [10]. Among these, CoronaVac^®^ is an inactivated vaccine against SARS-CoV-2 developed in Vero cells (Sinovac Life Sciences, Beijing, China). Preclinical studies performed in mice, rats, and non-human primates demonstrated that this vaccine was immunogenic and induced anti-SARS-CoV-2-neutralizing antibodies [11]. Moreover, partial or complete protection against pneumonia after a viral challenge was shown in primates [11]. All these results led to human clinical trials. Phase I/II sequential clinical trials were performed, including 144 and 600 healthy adults aged 18 to 59 years, respectively [12]. Two doses (3 and 6 µg) and two vaccination schedules (two doses separated by either two or four weeks) were evaluated. The results demonstrated that this inactivated vaccine was well tolerated with mild local adverse events after two doses [12]. Although anti-SARS-CoV-2 antibody-neutralizing geometric mean titers (GMTs) were lower when compared to convalescent patients, the vaccine induced a significant humoral response with both doses and schedules. A phase I/II sequential clinical trial performed in healthy adults aged 60 years and older showed that CoronaVac^®^ was safe and well-tolerated in this particular population [13]. Moreover, the 3 µg dose in the elderly group induced anti-SARS-CoV-2 antibody-neutralizing titers similar to those observed in adults aged 18–59 years. All these findings led to the emergency use of CoronaVac^®^ in China, and supported the development of a phase III study to evaluate the efficacy of this inactivated vaccine [14].

Due to the availability of CoronaVac^®^ to the general public in Chile since January 2021, we adapted the initial placebo-controlled phase III clinical trial in adults to a non-inferiority clinical trial of two different immunization schedules, with the second dose administered either two (0–14) or four (0–28) weeks after the first one, with a planned 12 months of follow-up. This report includes the safety and efficacy of non-inferiority results acquired up to six months after the first dose.

## 2. Materials and Methods

### 2.1. Study Design and Participants

This trial (clinicaltrials.gov NCT04651790) was a multi-center, randomized clinical trial to evaluate two vaccination schedules of CoronaVac^®^, the Sinovac inactivated SARS-CoV-2 vaccine, in adults in Chile that included healthcare workers and community participants recruited at eight sites (six in the Metropolitan Region of Santiago and two in the Valparaiso Region). The study was approved by the sponsoring institution Ethical Committee (Comité Ético Científico Ciencias de la Salud UC, Pontificia Universidad Católica de Chile, ID 200708006), each Institutional Ethical Committee of the other sites (Comité Ético Científico Universidad de Los Andes, Comité Ético Científico Facultad de Medicina Clínica Alemana, Universidad del Desarrollo, Comité Ético Científico Hospital Clínico Félix Bulnes, Comité Ético Científico Servicio de Salud Valparaíso-San Antonio, and Comité Ético Científico Servicio de Salud Metropolitano Sur Oriente, Chile), and the Public Health Institute of Chile (ISP Chile, number N° 24204/20). This study was also conducted according to the current Tripartite Guidelines for Good Clinical Practice, the Declaration of Helsinki [15], and local regulations. An independent data and safety monitoring board reviewed the blinded safety and efficacy data.

Written informed consent was obtained from each participant before enrollment. After inclusion and exclusion criteria were met (a complete list of inclusion and exclusion criteria has been published previously [16]), participants were randomly assigned to one of two open-label vaccination schedules, with either 14- (0–14) or 28-day (0–28) intervals between doses in a 1:1 ratio.

### 2.2. Aims

The primary safety endpoint was to evaluate the frequency of AEs occurring during the first 7 days after each dose of the vaccine in each vaccination schedule. The secondary endpoint was to determine the occurrence of SAEs and events of special interest in both vaccination schedules during the entire study.

The primary non-inferiority efficacy endpoint was to evaluate and compare the protection against confirmed SARS-CoV-2 infection for two vaccination schedules, starting two weeks after the second dose. The non-inferiority of the 0–14 schedule over the 0–28 schedule was defined as a difference in the protection rate within a threshold of 15%. The secondary efficacy endpoint was to compare the vaccination schedules regarding hospitalized cases and deaths within the same period.

### 2.3. Procedures

Demographic information, comorbidities, concomitant medications, and nutritional status were registered at enrolment and registered in a paper case report form (CRF) and an electronic CRF (eCRF). Blood samples and nasopharyngeal swabs were obtained for all participants prior to immunization to evaluate past or current SARS-CoV-2 infection. A urine test was performed on all female participants to assess potential pregnancy, which was an exclusion criterion. Participants were inoculated with 3 µg (600SU) of Coronavac^®^ and then kept in observation for 60 min after each dose to evaluate possible adverse events (AEs). An immediate AE was defined as reporting the AE within this period. Then, participants, or their representative, if applicable, were instructed to register through a remote application any local or systemic solicited AEs for 7 days after each dose, and any other AEs or concomitant medications until 28 days after the second dose. Non-immediate AEs were defined as those occurring after the first 60 min after vaccination. Serious adverse events (SAEs), events of special interest (using the priority List of Events of Special Interest for COVID-19 vaccines by Brighton Collaboration) [17], relevant medications (immunosuppressive drugs, transfusions, and other vaccines), and symptoms of SARS-CoV-2 infection were collected throughout the entire study. The system sent daily reminders to all participants until day 28 after the second dose, and then weekly until the end of the study. The severity of the solicited AE was graded through a numeric scale of 1 to 4 based on the “Toxicity Grading Scale for Healthy Adult and Adolescent Volunteers Enrolled in Preventive Vaccine Clinical Trials” guide of the Food and Drug Administration of the United States (FDA) [18]. The severity of the unsolicited clinical AE was classified through a numeric scale of 1 to 5, based on the “Common Terminology Criteria for Adverse Events—Version 5.0” guide by the United States National Cancer Institute (NCI /NIH) [19]. The investigators determined a possible causal association between the AE and vaccination according to a classification adapted from the “Uppsala Monitoring Center” of the World Health Organization [20]. Personnel of the sites reviewed this information for accuracy and completeness and filled an AE or SAE form in the eCRF.

To determine the protection against confirmed SARS-CoV-2 infection for each vaccination schedule, participants were followed during the study to identify and register any SARS-CoV-2 infection (COVID-19). The definition of case surveillance for COVID-19 was stated by the WHO [21]. Participants were instructed to register in the remote application and notify the site via mail, message, or phone call when they presented at least one of the symptoms for two days (suspicious case definition met). In these cases, a SARS-CoV-2 RT-qPCR was performed. A second sample was collected in the case of a negative RT-qPCR with persisting symptoms, and then a new RT-qPCR was performed. The investigators closely monitored participants who met the confirmed COVID-19 case definition (at least one symptom and a positive RT-qPCR), recording symptoms, severity, start and end dates, therapies, complications, hospitalizations, admission to the ICU, use of mechanical ventilation, and outcome. The severity of the COVID-19 symptoms was classified in grades 1 to 4 based on the “Toxicity Grading Scale for Healthy Adult and Adolescent Volunteers Enrolled in Preventive Vaccine Clinical Trials” guidelines from the United States Food and Drug Administration (FDA) and the “Common Terminology Criteria for Adverse Events–Version 5.0” guide by the United States National Cancer Institute (NCI /NIH) [18,19]. The intensity of the condition was registered using a scale of clinical progression (score 0 to 10) based on the WHO guidelines [20].

### 2.4. Statistical Analyses

Baseline characteristics of patients were compared for the two schedules: categorical variables are expressed as counts and percentages while numerical variables are expressed with mean and standard deviation (SD). Categorical variables were analyzed with the chi-square test or Fisher’s exact test; differences in means were tested using the Student’s t-test; and significance level was set at a more rigorous level of 0.01. The percentage of subjects that presented each solicited AE within the first 7 days was obtained for each schedule. The length of the event is presented as the median and quantiles 10 and 90. Incidences of immediate and non-immediate AEs were registered. The numbers of simultaneous non-immediate AEs are expressed as the sum of different AEs and are shown as frequency and percentage by dose and schedule. Differences in the incidence of each AE by age were evaluated using chi-square or Fisher's exact test. COVID-19 incidence, including only cases occurring 14 days after the second dose, was determined for each schedule, and subgroups were defined by sociodemographic or clinical characteristics. COVID-19-free survival was estimated using Kaplan–Meier analysis, and schedule curve differences were assessed using the log-rank test. Cox’s regression was used to obtain age and gender-adjusted incidence rate ratios and their 95% confidence intervals (CI). The proportional hazards assumption was met. For safety and efficacy, we looked at the non-inferiority of the 0–14 schedule over the 0–28 schedule with a margin of 15%. Consequently, one-sided statistical tests were used where the rejection of the null hypothesis indicated the non-inferiority of the 0–14 schedule over the 0–28 schedule. All statistical analyses were performed using SPSS 17.0.

### 2.5. Role of the Funding Source

The funder had no role in study design, data collection, analysis, interpretation, or report writing.

## 3. Results

### 3.1. Demographics and Participants

Between 29 November 2020 and 9 April 2021, a total of 2302 participants were vaccinated with the first dose of CoronaVac^®^. Of these participants, 1090 were allocated to the 0–14 schedule and 1212 to the 0–28 schedule. Safety and efficacy data derived from participants up to October 2021 are reported here, with a median (min–max) follow-up of 6.5 (0.5–6.7) months for the 0–14 schedule and 6.9 (1.1–7.1) months for the 0–28 schedule. Safety information for the 7 days after the first and second doses is available for 2302 and 2212 participants, respectively. These data were included in the safety analysis. Moreover, 2205 participants had clinical information 14 days after the second dose and were included in the efficacy analysis (Figure 1).

Demographic characteristics and comorbidities of the population are shown in Table 1. In the 0–14 schedule, there were significantly (*p* < 0.01) higher percentages of participants aged 18–59 years and health workers with a lower BMI than in the 0–28 schedule.

### 3.2. Safety Parameters

During the first 60 min after vaccination, 1–2% of participants reported local pain at the administration site. The other AEs were recorded in even lower frequency (Appendix A). No anaphylactic reactions were observed. After this immediate period, a total of 882 local and 1919 systemic solicited AEs were reported upon the administration of the first dose. These AEs were reported in 32.1% and 41.5% of the vaccinated participants for the 0–14 and 0–28 schedules. A total of 867 local and 1395 systemic solicited AEs were reported after the second dose. These AEs were reported in 31.2% and 32.9% of the vaccinated participants for the 0–14 and 0–28 schedules, respectively (Figure 2 and Appendix A). The 0–14 schedule showed no inferiority to the 0–28 schedule (*p* < 0.0001) in terms of the frequency of AEs (Appendix A).

After the first dose, 67.9% of the participants did not report any local AEs, 26.6% reported only pain at the inoculation site, 0.9% reported muscle pain and induration, 0.8% reported pain and local pruritus, and 0.6% reported only pruritus. All other combinations were found in less than 0.5% of the participants. After the second dose, 68.8% of the participants did not report any AE, 24.7% reported only pain at the inoculation site, 1.4% reported pain and induration, and 0.8% reported muscle pain, induration, and pruritus. All other combinations were found in less than 0.6% of the participants. The majority of the participants who presented any local AE after each dose reported one or two AEs (Appendix A).

After each dose, the most frequent solicited systemic AEs were headache, fatigue, and myalgia, reported in 20–26%, 12–17%, and 11–14% of the participants, respectively. The remaining systemic AEs were reported in less than 10% of the vaccinated participants. Notably, minor allergic reactions and fever were reported by less than 2% and 1% of the vaccinated participants, respectively (Figure 2 and Appendix A). The number of simultaneous systemic AEs is shown in Appendix A. The majority of the participants who presented any systemic AE reported one or two simultaneous AEs. Most local and systemic AEs were mild, with 0.5% or fewer participants reporting grade 3 AEs after the first dose and 0.6% or less after the second dose. There were no reports of grade 4 AEs (Figure 2). The most frequent local and systemic AEs were resolved in a median of 2 days (Appendix A). When comparing by age group, older participants (≥60 years old) showed less incidences of AEs than did younger participants (18–59 years old) (Appendix A).

No vaccine-related SAEs occurred, and 60 non-vaccine-related SAEs were reported, including 3 deaths. There were two sudden deaths, one due to acute myocardial infarction one month after the second dose (male, between 50 and 60 years old) and the other in a patient with a history of hepatic cirrhosis due to alcoholic liver disease, three months after the second dose (male between 60 and 70 years old). The third death was due to gastric cancer stage IV diagnosed five months after enrolment (female, between 70 and 80 years old). Five pregnancies were reported in participants of the study, two of them during the first four weeks after the second dose (one twin and one single pregnancy). All had a negative pregnancy test and contraceptive use before each vaccine dose, and these participants are being followed-up by the investigators, with no obstetric nor perinatal complications reported to date. To date, one of the pregnancies concluded with the birth of healthy twins. No other events of special interest have occurred in the study.

### 3.3. Vaccination Schedule Non-Inferiority Evaluation

Fourteen days after the administration of the second dose of CoronaVac^®^, fifty-eight symptomatic and confirmed COVID-19 cases were registered. The demographic and clinical characteristics of these COVID-19 cases are shown in Table 2. The vast majority of these cases were mild (Score 2) (94.8%), and just two participants were hospitalized. The first one was a male, aged over 60 years, with a BMI of 28.0 (overweight) and arterial hypertension and bicuspid aorta. This participant exhibited COVID-19 symptoms 32 days after the second dose of a 0–28 schedule and was a confirmed close contact with a COVID-19 case. The participant developed atrial fibrillation and heart failure and required mechanical ventilation (Score 7) for 6 days and hospitalization for 20 days. The second participant was male, aged over 60 years with a BMI of 29.3 (over-weighted), and was in treatment for hypothyroidism. The second participant exhibited COVID-19 symptoms 122 days after the second dose of the 0–28 schedule, and had no close contact with other COVID-19 cases. The participant received oxygen by nasal cannula (score 5) for four days and was released after seven days of hospitalization. Both participants exhibited cough, dyspnea, and fatigue for more than seven weeks, but ultimately recovered.

A total of 34 and 24 cases of COVID-19 were registered in the 0–14 and 0–28 schedules, respectively (*p* = 0.083) (Table 2). Both schedules showed a high probability of being COVID-19-free: 96.7% (0–14) and 97.9% (0–28) (non-inferiority *p*-value < 0.001). A Kaplan–Meier analysis showed that a probability of 0.98 for being COVID-19-free was achieved at day 91 for the 0–14 schedule and at day 133 for the 0–28 schedule. Although the COVID-19 incidence rate showed a slightly higher curve for the 0–14 schedule than the 0–28 schedule, this difference was not statistically significant (log-rank test, *p*-value = 0.071) (Figure 3). The 0–14 schedule showed non-inferiority to the 0–28 schedule when comparing COVID-19 incidence in different subpopulations defined by demographic and clinical characteristics (Appendix A).

The incidence of COVID-19 cases tends to be higher in health care workers compared with the general population for both immunization schedules (for 0–14 schedule, cases presented in 4.5 v/s 2.4% and for 0–28 in 3.1 v/s 1.7%); but these differences were not statistically significant. Additionally, the infection rate tends to be lower in ≥60 years old participants, but the significance level set for this analysis was not achieved (*p* = 0.024). No statistical differences were observed in the frequency of COVID-19 cases between sex and comorbidities.

## 4. Discussion

This non-inferiority trial has demonstrated that the virus-inactivated vaccine CoronaVac^®^, given in two doses with a 14- or 28-day interval between each dose, is safe, well-tolerated, and protective. A six-month surveillance showed the non-inferiority of the 0–14 schedule over the 0–28 schedule in solicited AEs and confirmed COVID-19 cases. These results further support the safety and protective capacity for the massive use of the Coronavac^®^ vaccine in adults, including participants older than 60 years old.

Regarding the safety of CoronaVac^®^, no vaccine-associated SAEs nor events of special interest were reported up until six months of follow-up in this cohort of over 2300 adults. During phase III trials with adenovirus and mRNA-based vaccine formulations, four and one SAEs were associated with each vaccine, respectively [22,23]. However, post-approval reports showed an increased incidence of vaccine-induced thrombotic thrombocytopenia, particularly for the adenovirus-based vaccine prototypes [23]. Moreover, cases of vaccine-related myocarditis were observed in adolescents and young adults vaccinated with mRNA-based vaccines [24].

CoronaVac^®^ shows a low reactogenicity profile, with around 30% of vaccinated participants reporting local pain, less than 1% fever, and no significant allergic reactions. In this line, the AEs reported in the phase III trials with CoronaVac^®^ performed in Turkey and Brazil were primarily mild and self-limited [25,26]. The low reactogenicity profile of CoronaVac^®^ contrasts with the relatively high incidence of local and systemic AEs reported post-vaccination for other vaccine platforms, such as mRNA and adenoviral vectors, with pain observed in 50–80%, fever in 16–51%, fatigue in up to 70%, and myalgias in up to 60% of the participants [22,27].

Regarding the vulnerable population, 30% of the enrolled participants in this study were ≥60 years old, and 45% had chronic conditions. Lower frequencies of post-vaccination AEs were observed for the older age cohort compared to the younger participants. Consistently with this notion, only one elderly subject developed a fever after vaccination, a condition that could escalate in older people. Concordantly, in a nationwide cross-sectional study for side effects of CoronaVac^®^ performed in Turkey, younger age was a risk factor associated with a discrete increase in vaccine side effects [26]. These features contribute to the confidence in the massive administration of this vaccine, especially in the most vulnerable populations.

Although the study design did not allow the calculation of true efficacy for the vaccine due to the absence of a placebo arm, we demonstrate that in a scenario of high viral circulation [3], the vast majority of COVID-19 cases developed by vaccinated participants were only mild. Only two participants aged over 60 years required hospitalizations, and no deaths due to COVID-19 occurred as part of the study [28]. Accordingly, we have previously published seroconversion data for COVID-19-infected participants showing high titers of anti-RBD antibodies [28]. These data are consistent with the immunogenicity results reported in Chile so far, showing that CoronaVac^®^ induces the secretion of specific IgG against S1-RBD with neutralizing capacity, as well as the activation of T cells specific to SARS-CoV-2 antigens [16]. Moreover, the wide use of this vaccine in the Chilean population has been monitored by the Ministry of Health and has shown an effectiveness of 67.7% to prevent symptomatic COVID-19 cases, and more than 85% to prevent severe COVID-19 cases and death due to SARS-CoV-2 infection [29]. Regarding the infected participants, the first cases reported were due to the delta strain, while the latest cases reported are mostly due to the omicron strain, which is in line with the prevalent strains reported in the corresponding dates.

Comparing the protective efficacy of two different vaccination schedules (0–14 vs. 0–28) against SARS-CoV-2 infection could help health authorities make evidence-based decisions for massive immunization against COVID-19. A more rapid schedule could lead to the faster vaccination of the population, which could be relevant during an epidemic. It is essential to evaluate differences regarding immunogenicity, efficacy, and effectiveness between an accelerated schedule versus a standard four-week interval. Two previous reports with this vaccine showed a more robust immune response for the 0–28 schedule than for the 0–14 schedule [12]. A phase I/II trial held in China showed higher neutralizing antibody seroconversion rates for the 0–28 schedule compared to the 0–14 schedule [12].

Regarding the non-inferiority evaluation, although we observed a trend towards a lower percentage of COVID-19 cases for the 0-28 schedule compared to the 0–14 schedule, these differences were not statistically significant. A previous study that evaluated the efficacy of CoronaVac^®^ in a 0–14 schedule demonstrated that this parameter was higher in participants who received the two doses with an interval of over 21 days [25]. An explanation for this apparent discrepancy is that, in our study, the group adhering to the 0–14 schedule consisted mainly of healthcare workers, who are usually more exposed to the virus and therefore have a higher risk of infection. Further studies with a more homogeneous population could contribute to addressing these questions.

Older age is a described risk factor related to COVID-19 severity [30], and this was also observed in our study. Here, the two severe cases reported occurred just in older participants. However, the frequency of cases tended to be lower in this age group. This could be related to the strictest protective measures taken in this population and their lower mobility during the time of the study.

After a six-month follow-up, two doses of CoronaVac^®^ were found to be well-tolerated, safe, and protective, particularly in a high-risk population. Regarding vaccination schedules, our data suggest that both a 0–14 and a 0–28 schedule show equivalent safety and efficacy results for this vaccine.

## Figures and Tables

**Figure 1 vaccines-10-01082-f001:**
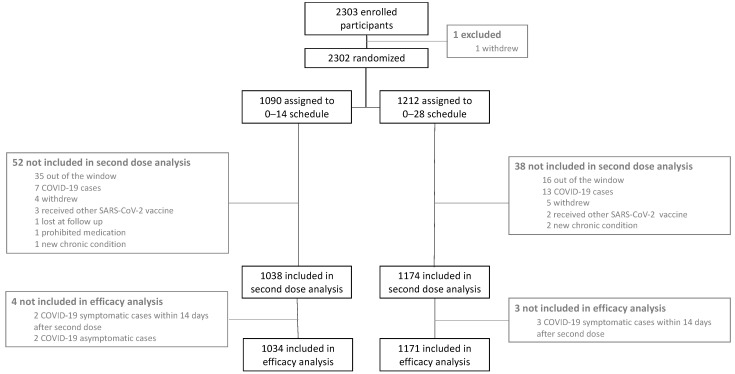
Study design for this phase III trial with two different immunization schedules as of October 2021. This study aims to characterize the safety and efficacy elicited by two immunization schedules with CoronaVac^®^, with each dose separated by either 14 or 28 days.

**Figure 2 vaccines-10-01082-f002:**
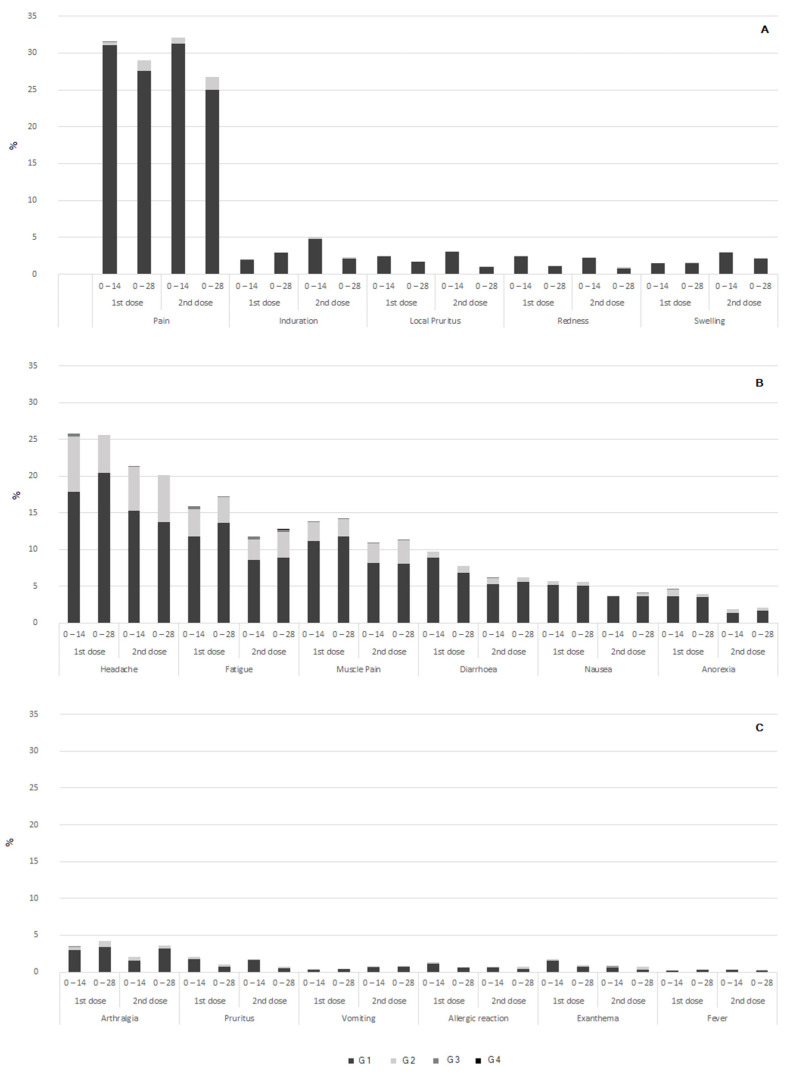
Frequency and severity of local and systemic adverse events by schedule and dose. Frequencies and severity grades are shown in percentages. (**A**) Local adverse events; (**B**,**C**) systemic adverse events.

**Figure 3 vaccines-10-01082-f003:**
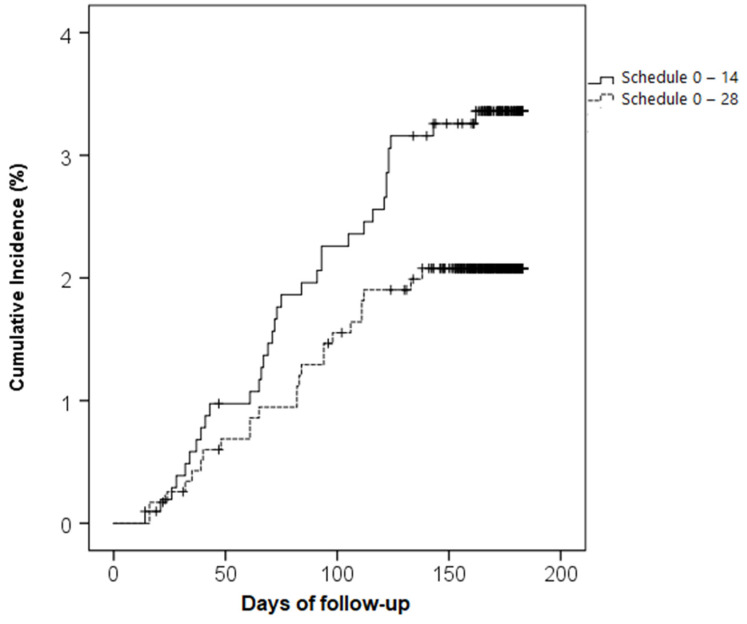
Cumulative incidences of COVID-19 infection by immunization schedule. Cumulative incidences of COVID-19 infection 14 days after administration of the second dose in two vaccination schedules (0–14 (solid line) and 0–28 (dashed line)). The X-axis shows days elapsed from the second dose to the event or censoring time. Censoring was set at the date of the third vaccination, retirement from the study, or reaching six months after the second dose, whichever occurred first.

**Table 1 vaccines-10-01082-t001:** Clinical and demographic characteristics of the study population at baseline.

	Total	Schedule 0–14	Schedule 0–28	*p* Value (a)
	(*n* = 2302)	(*n* = 1090)	(*n* = 1212)	
Age; n (%)				**0.001**
18–59	1616 (70.2)	800 (73.4)	816 (67.3)	
60–98	686 (29.8)	290 (26.6)	396 (32.7)	
Sex; n (%)				0.015
Female	1212 (52.6)	603 (55.3)	609 (50.2)	
BMI; mean ± SD	26.9 ± 4.5	26.6 ± 4.4	27.1 ± 4.6	**0.003**
Ethnicity; n (%)				0.128
Hispanic or Latino	2294 (99.7)	1083 (99.4)	1211 (99.9)	
Chilean native	4 (0.2)	3 (0.3)	1 (0.1)	
Asian	3 (0.1)	3 (0.3)	0 (0.0)	
Black	1 (0.0)	1 (0.1)	0 (0.0)	
Health workers; n (%)				**<0.001**
Yes	759 (33.0)	459 (42.1)	300 (24.8)	
Comorbidities; n (%)				
≥1	1042 (45.3)	487 (44.7)	555 (45.8)	0.580
Comorbidities; n (%)				
Cardiovascular disease	34 (1.5)	14 (1.3)	20 (1.7)	0.468
Asthma and COPD *	148 (6.4)	83 (7.6)	65 (5.4)	0.028
Diabetes	117 (5.1)	44 (4.0)	73 (6.0)	0.030
Insulin resistance	180 (7.8)	85 (7.8)	95 (7.8)	0.967
Hypothyroidism	248 (10.8)	135 (12.4)	113 (9.3)	0.018
Arterial hypertension	415 (18.0)	174 (16.0)	241 (19.9)	0.015
Allergic rhinitis	300 (13.0)	137 (12.6)	163 (13.4)	0.531
Thyroid disease	251 (10.9)	136 (12.5)	115 (9.5)	0.022
Obesity	478 (20.8)	205 (18.8)	273 (22.5)	0.028
Dyslipidaemia	43 (1.9)	18 (1.7)	25 (2.1)	0.467

**Bold***p*-values were considered statistically significant. *** COPD**: Chronic obstructive pulmonary disease.

**Table 2 vaccines-10-01082-t002:** Characteristics of COVID-19-positive participants by the immunization schedules.

	Total	Schedule 0–14	Schedule 0–28
	(*n* = 58)	(*n* = 34)	(*n* = 24)
Age in years, n (%)			
18–59	48 (82.8)	29 (85.3)	19 (79.2)
60–98	10 (17.2)	5 (14.7)	5 (20.8)
Sex, n (%)			
Female	31 (53.4)	18 (52.9)	13 (54.2)
Clinical score, n (%)			
2 (symptomatic, independent)	55 (94.8)	33 (97.1)	22 (91.7)
3 (symptomatic, assistance needed)	1 (1.7)	0 (0.0)	1 (4.1)
5 (hospitalized, oxygen by mask or nasal prongs)	1 (1.7)	1 (2.9)	0 (0.0)
7 (intubation and mechanical ventilation)	1 (1.7)	0 (0.0)	1 (4.1)
Severity criteria, n (%)			
Hospitalizations	2 (3.4)	1 (2.9)	1 (4.1)
UCI admissions	1 (1.7)	0 (0.0)	1 (4.1)
Deaths	0 (0.0)	0 (0.0)	0 (0.0)
Health setting workers, n (%)			
Yes	28 (48.3)	19 (55.9)	9 (37.5)
Comorbidities, n (%)			
≥1	26 (44.8)	14 (41.2)	12 (50.0)
Comorbidities, n (%)			
Cardiovascular disease	4 (6.9)	2 (5.9)	2 (8.3)
Asthma and COPD *	6 (10.3)	3 (8.8)	3 (12.5)
Diabetes	3 (5.2)	2 (5.9)	1 (4.2)
Insulin resistance	5 (8.6)	1 (2.9)	4 (16.7)
Arterial hypertension	14 (24.1)	9 (26.5)	5 (20.8)
Allergic rhinitis	7 (12.1)	3 (8.8)	4 (16.7)
Thyroid disease	3 (5.2)	3 (8.8)	0 (0.0)
Obesity	14 (24.1)	5 (14.7)	9 (37.5)
Dyslipidaemia	0 (0.0)	0 (0.0)	0 (0.0)

*** COPD**: chronic obstructive pulmonary disease. Data are presented as frequency and percentage of the total number of cases in each subgroup and were compared with chi-square test or Fisher exact test; all *p* values were higher than 0.05.

## Data Availability

All analyzed and raw data (masked to protect the information of volunteers) are available upon reasonable request to the corresponding authors through email after the publication of this article. A signed data access agreement will be requested to share the data. The study protocol is also available online and annexed to this article.

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
