# Peer review of "Safety and Non-Inferiority Evaluation of Two Immunization Schedules with an Inactivated SARS-CoV-2 Vaccine in Adults: A Randomized Clinical Trial"

_vaccines, 2022, doi:10.3390/vaccines10071082_

Round 1

Reviewer 1 Report

The paper is interesting and well presented.

Only one comment for the explanation presented on lines 212-215 about the information presented in Table 1. The text  indicates that there are statistical differences only for health workers and BMI, but statistical differences (p < 0.05) are also present fot age, sex, hypertension, asthma, hypertension and obesity. This must be clarified.

Author Response

Reviewer: The paper is interesting and well presented. Only one comment for the explanation presented on lines 212-215 about the information presented in Table 1. The text indicates that there are statistical differences only for health workers and BMI, but statistical differences (p < 0.05) are also present for age, sex, hypertension, asthma, hypertension and obesity. This must be clarified.

Answer: We would like to kindly thank the Reviewers for their comments. As it was indicated in the Statistics Analyses Section in the Material and Methods section, we decided to set a significance level of 0.01 throughout our analyses as a stricter cut off (Lines 182-182).

We would like to thank once again the Reviewers and the Editors for their time and effort in the handling of this document, and we hope that the current revised manuscript is acceptable for publication in Vaccines.

Reviewer 2 Report

The paper reports the results of a comparison study of two vaccination schedules with CoronaVac, an inactivated virus-based anti- SARS-CoV2 vaccine, administered as two doses of 3 ug  either 14 or 28 days apart. The primary objective was to evaluate the safety of the vaccine and the non-inferiority hypothesis of either of the vaccination schedules.  The data analyzed are from 1090 (after the 1st dose) and 1034 (after the 2nd dose) subjects form the 14 day interval vaccination schedule, and 1212 (after the 1st dose) and 1174 (after the 2nd dose) subjects from the 28 days vaccination schedules, respectively. The data demonstrate the prevalence of mild adverse events shortly after the vaccine administration, which were similar between both vaccination schedules. The non-inferiority analysis demonstrated no statistically significant differences in the frequency or the severity of confirmed SARS-CoV2 cases between both vaccinated groups. These data are analyzed correctly and clearly presented.

However, the claim of the “efficacy” of the vaccination schedules cannot be substantiated by this study since a placebo group is absent, and neither the screening of the enrolled subjects for the presence of anti-SARS-CoV2 antibodies nor ab analysis after the vaccination were performed. The only support for the “efficacy” is drawn from the number of cases (58) of confirmed SARS-CoV2 in both vaccinated groups, which is meaningless without a placebo comparison group.

The corresponding statements about “efficacy” should be removed from the Abstract (lines 47-48), and Discussion (lines 373-374). The Results section 3.3 “vaccine efficacy” should be renamed to “vaccination schedule non-inferiority evaluation”

Author Response

Reviewer: The paper reports the results of a comparison study of two vaccination schedules with CoronaVac, an inactivated virus-based anti- SARS-CoV2 vaccine, administered as two doses of 3 ug either 14 or 28 days apart. The primary objective was to evaluate the safety of the vaccine and the non-inferiority hypothesis of either of the vaccination schedules. The data analyzed are from 1090 (after the 1st dose) and 1034 (after the 2nd dose) subjects form the 14 day interval vaccination schedule, and 1212 (after the 1st dose) and 1174 (after the 2nd dose) subjects from the 28 days vaccination schedules, respectively. The data demonstrate the prevalence of mild adverse events shortly after the vaccine administration, which were similar between both vaccination schedules. The non-inferiority analysis demonstrated no statistically significant differences in the frequency or the severity of confirmed SARS-CoV2 cases between both vaccinated groups. These data are analyzed correctly and clearly presented. However, the claim of the “efficacy” of the vaccination schedules cannot be substantiated by this study since a placebo group is absent, and neither the screening of the enrolled subjects for the presence of anti-SARS-CoV2 antibodies nor ab analysis after the vaccination were performed. The only support for the “efficacy” is drawn from the number of cases (58) of confirmed SARS-CoV2 in both vaccinated groups, which is meaningless without a placebo comparison group. The corresponding statements about “efficacy” should be removed from the Abstract (lines 47-48), and Discussion (lines 373-374). The Results section 3.3 “vaccine efficacy” should be renamed to “vaccination schedule non-inferiority evaluation”

Answer: We would like to kindly thank the Reviewers for their comments. As requested, we have modified the manuscript to reflect the changes indicated (Lines 2, 37, 47, 274, 372-374).

We would like to thank once again the Reviewers and the Editors for their time and effort in the handling of this document, and we hope that the current revised manuscript is acceptable for publication in Vaccines.

Reviewer 3 Report

Although several vaccines against COVID-19 are available, the development of classical inactivated vaccine is considered as important due to lower reactogenicity. The clinical trial of the CoronaVac exhibited a promissing outcome so far. Only a few points were raised.

1. Do authors have any seroconversion data, particularly for the COVID-19 infected participants ?

2. Do authors have any information on the infected virus variants of the COVID-19 infected participants ?

3. Line 226: Figure1 should read as Figure 2

4. Line 238: The term "muscle pain " is used in the figure. Please correct either one.

5. To describe clinical trial phase, authors used "1, 2, 3" and "I, II, III". Please use one of them.

Author Response

Reviewer: Although several vaccines against COVID-19 are available, the development of classical inactivated vaccine is considered as important due to lower reactogenicity. The clinical trial of the CoronaVac exhibited a promising outcome so far. Only a few points were raised. 1. Do authors have any seroconversion data, particularly for the COVID-19 infected participants?

Answer: As requested by the Reviewer, for the COVID-19 infected volunteers, these data were already published previously, and the corresponding reference was included (Lines 351-353).

Reviewer: Do authors have any information on the infected virus variants of the COVID-19 infected participants?

Answer: As requested by the Reviewer, we added information regarding the variants of SARS-CoV-2 in the infected participants (Lines 359-362).

Reviewer: Line 226: Figure1 should read as Figure 2

Answer: As requested by the Reviewer, we have corrected this issue in our manuscript (Line 231).

Reviewer: Line 238: The term "muscle pain " is used in the figure. Please correct either one.

Answer: As requested by the Reviewer, we updated the manuscript to reflect this information (Lines 235 and 239).

Reviewer: To describe clinical trial phase, authors used "1, 2, 3" and "I, II, III". Please use one of them.

Answer: As requested by the Reviewer, we have updated this terminology throughout the manuscript.

We would like to thank once again the Reviewers and the Editors for their time and effort in the handling of this document, and we hope that the current revised manuscript is acceptable for publication in Vaccines.